# Molecular Biomarkers for Pediatric Depressive Disorders: A Narrative Review

**DOI:** 10.3390/ijms221810051

**Published:** 2021-09-17

**Authors:** Jongha Lee, Suhyuk Chi, Moon-Soo Lee

**Affiliations:** 1Department of Psychiatry, Korea University Ansan Hospital, Ansan 15355, Korea; purified0106@hotmail.com; 2Department of Psychiatry, Korea University Guro Hospital, Seoul 08308, Korea; link0710@hotmail.com

**Keywords:** pediatric, depression, biomarker, BDNF, cytokines

## Abstract

Depressive disorder in childhood and adolescence is a highly prevalent mood disorder that tends to recur throughout life. Untreated mood disorders can adversely impact a patient’s quality of life and cause socioeconomic loss. Thus, an accurate diagnosis and appropriate treatment is crucial. However, until now, diagnoses and treatments were conducted according to clinical symptoms. Objective and biological validation is lacking. This may result in a poor outcome for patients with depressive disorder. Research has been conducted to identify the biomarkers that are related to depressive disorder. Cumulative evidence has revealed that certain immunologic biomarkers including brain-derived neurotrophic factor (BDNF) and cytokines, gastrointestinal biomarkers, hormones, oxidative stress, and certain hypothalamus-pituitary axis biomarkers are associated with depressive disorder. This article reviews the biomarkers related to the diagnosis and treatment of pediatric depressive disorders. To date, clinical biomarker tests are not yet available for diagnosis or for the prediction of treatment prognosis. However, cytokines such as Interleukin-2, interferon-gamma, tumor necrosis factor-alpha, and BDNF have shown significant results in previous studies of pediatric depressive disorder. These biomarkers have the potential to be used for diagnosis, prognostic assessment, and group screening for those at high risk.

## 1. Introduction

Depressive disorder, one of the most common psychiatric disorders in children and adolescents, negatively affects an individual’s psychosocial development [1]. Pediatric depression can cause poor academic achievement and interpersonal relationships, and is highly related to suicidal behaviors [2,3]. Despite the growing interest in pediatric depression, the prevalence of major depressive episodes among adolescents has only gradually increased, and in 2014, 11.3% of American adolescents reported experiencing major depressive episodes [4]. Because untreated depressive disorder in children and adolescents persists into adulthood and negatively affects quality of life, it is very important to screen at-risk groups for depression and treat adolescents with depressive disorder in the early phase. The diagnosis of depressive disorder is made through a clinician’s evaluation according to the individual’s clinical symptoms, since objective diagnostic evaluations are lacking. Biomarkers that can screen an individual for the risk of developing a depressive disorder are lacking, which delays the treatment intervention of adolescents with early depressive disorder. In addition, the biological predictors of responses to clinically available treatments have not yet been determined. More than half of patients do not achieve remission during their first course of treatment [5,6]. Efforts have been made to explore the risk factors and etiologies of depression and improve upon objective diagnostic methods. These characteristics, which are objectively measured and evaluated as indicators of normal biological processes, pathogenic processes, or pharmacological responses to therapeutic interventions, are termed biomarkers [7].

Previous studies of various biomarkers have been conducted in adults with depression. Similarly, studies have been conducted on biomarkers related to the diagnosis of depression in children and adolescents, but the study results are not necessarily consistent with those of adults. Children and adolescents are in a developmental stage unlike adults, and the hormonal system of adolescents is also different. In addition, studies on adults with depressive disorder have a limited ability to clarify the number and duration of depressive episodes and the onset of depressive symptoms, making it difficult to identify changes in the early stages. It is challenging to rule out the impact of psychiatric drugs, such as antidepressants and antipsychotics, in adult depressive disorder. It is also problematic to clarify the effect of physical health on the biomarkers in adult depressive disorder. It is unclear whether the increased inflammation observed in major depressive disorder (MDD) is its cause or result [8]. There seems to be also some differences between pediatric and adult depression due to (1) developmental changes, (2) course of illness factors (e.g., number of episodes, chronicity, and total duration of illness), and (3) heterogeneity in clinical outcomes (e.g., transition into bipolar disorder) [9]. To minimize the influence of other variables affecting MDD, it is necessary to search for biomarkers for affected children and adolescents. Here, we aim to review the biomarkers related to pediatric depression to identify risk factors and early changes in depression. We review the neuroimmune system, neurohormonal system, gastrointestinal (GI) biomarkers, and oxidative stress biomarkers in pediatric depression and summarize the results that can be helpful in their assessment and clinical use. 

## 2. Neuroimmune System and Pediatric Depressive Disorder

Although inconsistent results have been reported, several cytokines have been suggested to be associated with depressive disorders. Relatively consistent findings showed an increase in several cytokines, including interleukin-6 (IL-6), interleukin-1 (IL-1), tumor necrosis factor-alpha (TNF-α), and C-reactive protein (CRP) in patients with depressive disorder versus healthy controls (HCs) [10,11,12,13,14,15]. An increase or decrease in certain cytokines also reportedly affects treatment outcomes with antidepressants [14,16,17]. In the association between the neuroimmune system and depressive disorder, interest has increased in the role of brain-derived neurotrophic factor (BDNF) as well as cytokines. BDNF is a neurotrophin that regulates neurogenesis, neuronal maturation, survival, and synaptic plasticity [18,19]. Several previous studies have suggested that BDNF affects neuroimmune regulation in psychiatric disorders, including schizophrenia, mood disorders, and obsessive disorder [20,21,22]. We discuss each in more detail in the following sections.

### 2.1. The Cytokines in Neuroinflammation

Neuroinflammation is defined as the reactive state of astrocytes and microglia, induced by pathological conditions [23]. These cells mediate the immune responses in the brain by producing and secreting pro-inflammatory cytokines that are known to be associated with depressive disorder [24,25]. The secretion of peripheral cytokines increases in pathologic states, such as infection, and affects hypothalamic-pituitary-adrenal (HPA) axis activation. Cytokines are a family of polypeptides that are important in cell signaling, and they affect the behavior of cells. Cytokines can access the brain by passing through leaky regions in the blood-brain barrier. They have functions such as neuroprotection and neurodegeneration in the central nervous system (CNS). They also have significant effects on neurotransmitters such as dopamine and serotonin [26,27]. Mae et al. showed that the activity of IL-6 was higher in patients with depressive disorder than in controls. In the same study, increased IL-6 activity was related to hyperactivity of the HPA axis as well as high plasma cortisol levels [16,28]. Since then, the number of studies on the relationship between cytokines and depressive disorder has increased. These studies have investigated several markers including IL-1, IL-1β, IL-6, IL-8, IL-10, TNF-α, interferon-γ (IFN-γ), and CRP [29]. Cytokines are generally divided into pro-inflammatory and anti-inflammatory cytokines [30]. Pro-inflammatory cytokines, including IL-1β, IL-6, IL-12, TNF-α, and IFN-γ, are thought to be associated with depressive disorders [11,14,15,17,27,31,32,33]. Additionally, IL-6 and TNF-α negatively impact serotonin production and integrity [34], which may increase the risk of depression. Furthermore, the release of TNF-α, as well as IL-1β, is thought to induce synaptic pruning. This leads to impaired neuroplasticity and structural brain changes that negatively impact cognition [34,35]. CRP is an acute-phase protein that is released in response to inflammation and increases cytokine levels [36,37]. Evidence of an association between inflammatory cytokines and the pathophysiology of depressive disorder has been reported; however, the direction of this association remains unclear. Table 1 summarizes the previous studies of cytokines in pediatric depressive disorder.

Cross-sectional and longitudinal studies have been conducted on the relationship between pediatric depression and inflammatory markers [10,11,12,13,14,17,38,39]. The cytokines that have been studied in pediatric depression are IL-1, IL-1β, IL-2, IL-6, IL-10, TNF-α, and IFN-γ. Gabbay et al. measured the plasma levels of IL-1β, IL-6, IL-10, TNF-α, and IFN-γ in adolescents with depressive disorder as well as in HCs [10]. This study showed that plasma IFN-γ levels were significantly higher in the patient group than in the healthy control group. The same results were also found in non-medicated patients with MDD compared to HCs. Increased plasma IL-6 levels are among the most consistent findings in depression [33,40,41]. Similar results were reported by a study of 42 female adolescents with depression, conducted by Blom et al. [11]. Plasma levels of IL-6 and IFN-γ are associated with depression and anxiety symptoms’ severity. IL-6 levels were higher in the non-medicated patient group than in the medicated patient group. In addition to IL-6 and IFN-γ, IL-2, IL-10, and IL1-β levels were significantly higher in the patient group. In a meta-analysis of 22 studies (20,791 participants) of the association between depressive symptoms/depressive disorder and inflammation in children and adolescents, IL-6 level was correlated with depressive symptoms. Elevated IL-6 levels were evaluated as predictors of future depression [41,42]. Evidence has also been gathered on the association between IL-6 level and internalizing disorder symptoms [35,43]. In a study of 134 students (n = 76 with internalizing disorder, n = 58 without internalizing disorder) aged 10–17 years conducted by Cristiano Tschiedel Belem da Silva and colleagues in Brazil, students with internalizing disorders, including MDD, generalized anxiety disorder, separation anxiety disorder, social anxiety disorder, or panic disorder showed significantly higher IL-6 levels than students without an internalizing disorder [12]. Internalizing behaviors evaluated at 8 years of age were associated with elevated IL-6 levels measured at 10 years of age in a large community cohort in England [43]. Although there were no consistent results in pediatric depression, IL-1β levels were significantly associated with pediatric depression, and higher in MDD patients than in HCs [11,13,33]. Pallavi et al. found that plasma levels of IL-1 β did not differ significantly between patients with MDD and controls. However, IL-1β levels are associated with anxiety symptoms’ severity in patients with MDD [33]. In addition, the IL-1β level has been suggested to be related to treatment-refractory depression. Amitai et al. evaluated the plasma levels of IL-1β, TNF-α, and IL-6 in 41 adolescents aged 9–12 years with depressive and/or anxiety disorders [17]. IL-1β, TNF-α, and IL-6 levels were high in the selective serotonin reuptake inhibitor (SSRI)-refractory patient group. This suggests the possible predictability of resistance to fluoxetine treatment in children and adolescents.

TNF-α is also reportedly highly relevant in pediatric depression. In previous studies of adult and pediatric depression, plasma levels of TNF-α levels showed mixed results. However, treatment with cytokines, including TNF-α, reportedly induces depressive symptoms [44,45,46]. Plasma levels of TNF-α did not differ significantly or were increased in patients with MDD versus controls [10,11,13,14,38,39]. However, a decrease in plasma TNF-α levels after antidepressant treatment in patients with MDD was reported [13,17]. In these studies, a statistically significant decrease in plasma TNF-α levels was reported 4 and 8 weeks after antidepressant drug treatment, respectively. Studies have examined the relationship between TNF-α level and childhood trauma. The relationship between childhood trauma and blood TNF-α level is reportedly unclear [47,48]. Peters et al. reported that TNF-α was associated with reduced inhibitory control performance in adolescents with depression and childhood trauma [38]. Rengasamy et al. suggested that higher baseline levels of TNF-α are associated with greater depression and anhedonia symptoms’ severity. They found that a higher baseline TNF-α level affects the depressive trajectory [39]. In a meta-analysis conducted by D’Acunto et al. on pediatric depression (despite including a small number of studies), the mean TNF-α level was higher in the MDD patient group than in the healthy control group [49]. On the other hand, Gabbay et al. reported that suicidal adolescents with MDD showed lower plasma TNF-α levels than non-suicidal adolescents with MDD. Therefore, further research on TNF-α level is required [32]. The relationship between CRP and pediatric depression has not been clearly identified, and contradictory results have been reported. Chaiton et al. found no significant association between depressive symptoms and serum CRP levels after the adjustment for variables such as body mass index, smoking, and blood pressure [50]. In a study by Copeland et al., CRP levels did not predict later depression. However, cumulative depressive episodes were suggested to affect later CRP levels [37]. On the other hand, Miller and Cole reported that the transition to depression in adolescents previously exposed to childhood adversity was accompanied by increased CRP levels [51]. In this study, CRP levels remained high in adolescents with depression and childhood adversity, even 6 months after the depressive symptoms improved. This suggests a relationship between childhood adversity and the neuroinflammatory system. In a study of Iranian female adolescents, Tabatabaeizadeh et al. showed a higher mean CRP level in the depressive disorder group, while serum CRP levels were positively associated with depressive symptoms’ severity [52]. In a meta-analysis by Colasanto et al., a significant association between CRP levels and depression was observed, although causality could not be inferred [41].

Similar to the previous cytokine study of adults, increased levels of IL-6 were found in cases of pediatric depression and decreased after treatment. In addition, increased IL-6 levels in pediatric depression are associated with adverse childhood experiences (ACEs). In a previous study, depressed patients with ACEs had increased IL-6 levels compared to depressed patients without ACEs in adults, which suggests the possibility that an increased IL-6 level may be a change seen in the MDD risk group and patients with early-stage MDD [53]. TNF-α levels were increased in cases of pediatric and adult depression, but the response to treatment differed. Although TNF-α tends to decrease after medication treatment in pediatric depression, an increase in TNF-α level was observed after treatment in adult depression [54]. It is possible that the underlying pathophysiology of depression differs between adolescents and adults, and future studies controlling other variables such as medication exposure and ACEs are needed.

**Table 1 ijms-22-10051-t001:** Previous studies of cytokines in pediatric depressive disorder.

Study	Objective	Design	Inflammatory Markers	Findings
Gabbay et al., 2009 [10]	To examine the immune system in adolescents with MDD	N: 45, age 12–19 years; 13 psychotropic-free MDD Pts, 17 MDD Pts with medication, and 15 HCs	Plasma IL-6, IFN-γ, TNF-α, IL-4, and IL-1β	Significantly higher levels of plasma IFN-γ in MDD Pts and trend for IL-6 to be elevated in MDD group; Significantly increased level of IFN-γ in the unmedicated MDD group compared to HCs
Henje Blom et al., 2011 [11]	To investigate the effects of antidepressants on cytokines in adolescent females with anxiety disorder and/or depressive disorder	N: 102, age 14.5–18.4 years; 42 Pts (26 unmedicated Pts and 16 SSRI Pts) and 60 HCs	Plasma IL-1β, IL-2, IL-6, IL-10, TNF-α, and IFN-γ	Significantly higher values of IL-2, IL-10, and IL1-β in patient group; higher level of IL-6 in the non-medicated subgroup compared to the medicated subgroup; higher levels of IL-6 and IFN-γ were significantly related to more severe self-assessed symptoms of anxiety and depression
Copeland et al., 2012 [37]	To test (1) the effect of CRP levels on later depression status; (2) the effect of depression status on later CRP levels; and (3) the effect of cumulative depressive episodes on later CRP levels.	N: 1420, age 9, 11, and 13 years at intake; longitudinal study with annual assessment to age 16 and again at 19 and 21 years	CRP (dried blood spot)	CRP levels were not associated with later depression status; Cumulative depressive episodes predicted later CRP levels
Rengasamy et al., 2012 [39]	To examine the associations of IL-6 and TNF-α with depression severity and anhedonia severity	N: 36, age 12–18 years; 36 adolescents with depressive disorder, cross-sectional and longitudinal study	TNF-α, and IL-6	Baseline TNFα was positively associated with baseline and follow-up SHAPS anhedonia scores, and follow-up CDRS-R
Amitai et al., 2016 [17]	To determine whether plasma levels of pro-inflammatory cytokines can predict response to treatment and/or are altered post fluoxetine treatment in children and adolescents.	N: 41, age 7–18 years; children and adolescents with depression and/or anxiety disorders.	Plasma IL-1β, IL-6, and TNF-α	Significantly higher levels of pro-inflammatory cytokines in SSRI-refractory than in SSRI-responsive Pts; TNF-α levels significantly reduced after 8 weeks of antidepressant treatment
da Silva et al., 2017 [12]	To compare serum levels of IL-6 and IL-10 between non-medicated adolescents with internalizing disorders and a comparison group of adolescents without internalizing disorders	N: 134, age 10–17 years; 76 adolescents with internalizing disorder and 58 adolescents without internalizing disorder	Plasma IL-6, and IL-10	Adolescents with internalizing disorders had significantly higher levels of IL-6 as compared to those without internalizing disorders
Pérez-Sánchez et al., 2018 [13]	To detect (1) the alterations in the cytokine profiles of adolescents during 8 weeks of treatment with fluoxetine and (2) the correlation between symptomatology and inflammatory profiles	N: 40, age 14–19 years; 22 adolescents with first episode of MDD and 18 HCs, cross-sectional and longitudinal study	Plasma IL-2, IFN-γ, IL-1β, TNF-α, IL-6, IL-15, IL-10, IL-5, IL-13, IL-1Ra, and IL-12p70	Significantly increased levels of pro-inflammatory cytokines (IL-2, IFN-γ, IL-1β, TNF-α, IL-6, IL-12, and IL-15) and anti-inflammatory cytokines (IL-4, IL-5, and IL-13) in MDD Pts; IFN-γ, IL-1β, TNF-α, IL-6, IL-12, and IL-15 decreased only at week 4; increased IL-2 only at week 8; increased anti-inflammatory cytokines IL-4 and IL-5 at week 8
Peters et al., 2019 [38]	1. To compare groups with inflammation 2. To evaluate associations between inflammation and inhibitory control	N: 70, age 12–17 years; 22 depressive adolescents with childhood trauma (DEP-T), 18 depressive adolescents (DEP), and 30 HCs	Plasma IL-1β, TNF-α, IL-6	Significantly elevated levels of IL-6 in both DEP and DEP-T relative to HCs and significantly elevated levels of TNF-α in DEP; No group differences were detected in IL-1β; TNF-α was associated with behavior-based and observer-rated inhibitory control deficits
Lee et al., 2020 [14]	1. To examine (1) the difference between inflammatory markers in MDD Pts and HCs and (2) whether these changes would be altered following antidepressant treatment 2. To investigate the relationship between cytokines’ level with the severity of depression	N: 50, age 13–18 years; 25 medication-naïve MDD Pts and 25 HCs, cross-sectional and longitudinal study	Plasma IL-1β, IL-2, IL-4, IL-6, IL-10, TNF-α, and IFN-γ	MDD Pts had significantly decreased level of plasm IL-2, IFN-γ, TNF-α, and IL-10 compared to healthy controls; IL-2, IFN-γ, and IL-10 showed significant increases after 12 weeks treatment compared to before treatment. IFN-γ level was negatively correlated with the CDI (r = −0.377, *p* < 0.01) and HDRS score (r = −0.457, *p* < 0.01)

Self-report Snaith−Hamilton Pleasure Scale (SHAPS), Children’s Depression Rating Scale-Revised (CDRS-R), Healthy controls (HCs), Patients (Pts).

### 2.2. BDNF in the Neuroimmune System

BDNF is a protein of the neurotrophin family and is encoded by the BDNF gene. Previous studies have suggested a link between BDNF and depressive disorders [55,56,57,58]. BDNF is synthesized in the brain and is widely distributed in the CNS [59]. BDNF plays an important role in neuronal development, neuroprotection, and the modulation of synaptic interactions and neuroimmune axis regulation [60]. BDNF influences neuronal structure and functional plasticity [20,60]. Neural plasticity dysfunction is related to the pathophysiology of depression. The BDNF hypothesis postulates that the loss of BDNF plays a major role in the pathophysiology of depression [58,61,62]. In addition, changes in BDNF levels were noted in patients with depressive disorder after appropriate treatment. This treatment included antidepressant medication, repetitive transcranial magnetic stimulation, and electroconvulsive therapy (ECT) [63,64,65,66]. In a meta-analysis, patients with depressive disorder showed increased BDNF levels after ECT. BDNF has been suggested as a potential indicator of the ECT response [65]. Low levels of BDNF have been suggested to be associated with suicidality in patients with depressive disorders [67,68]. The results of these changes in BDNF levels were relatively consistent in adults with MDD. However, in children and adolescents, the results were inconsistent [55,56,57,69]. Table 2 shows a summary of the previous studies of BDNF in pediatric depressive disorder.

Several studies of the relationship between pediatric depressive disorder and serum BDNF levels have suggested that the MDD group showed decreased BDNF levels compared to the healthy control group. This is similar to the results observed in adult depressive disorder. Pandey and colleagues observed decreased gene expression of BDNF in lymphocytes and decreased protein expression of BDNF in the platelets of adult and pediatric depressed patients versus HCs [21]. This study noted no significant relationship between BDNF levels and depressive symptom severity. This suggests that reduced BDNF levels may be associated with the diagnosis of depressive disorder. Sun et al. observed lower BDNF levels in patients with depressive disorder [69]. In this study, the group receiving comprehensive treatment showed higher serum BDNF levels than the group receiving routine treatment. The quality of life index showed greater improvement in the group that received comprehensive treatment. Studies reported that the role of BDNF differs between the sexes. In a Japanese study of adolescents aged 8–15 years, Sasaki et al. confirmed that the serum BDNF level differed according to sex in patients with MDD [55]. Male pediatric patients with MDD showed significantly decreased serum BDNF levels compared to male HCs; however, female pediatric patients with MDD did not. Furthermore, there was a significant negative correlation between serum BDNF levels and illness duration in men only. This study reported no correlation between serum BDNF levels and the Children’s Depression Rating Scale-Revised (CDRS-R) scores in pediatric patients with MDD. They suggested that decreased serum BDNF levels may play an important role in the pathophysiology of male pediatric depressive disorder [55]. Similar results of alterations in BDNF serum levels in adolescents were also confirmed by Pallavi et al. [56]. Both male and female adolescents with depressive disorder showed lower BDNF levels than HCs. However, BDNF levels were negatively correlated with depressive symptom severity only in male patients. BDNF expression and activity are affected by female hormones such as estrogen [70,71]. For this reason, the authors suggested that BDNF may play different roles in depressive disorders between the sexes [56]. 

One study showed conflicting results with respect to higher serum BDNF levels in adolescent depressive disorder. Bilgiç et al. compared 70 treatment-free patients with MDD to a healthy control group aged 11–19 years [57]. Serum BDNF levels were significantly higher in adolescents with depressive disorder, and there was no correlation between BDNF levels and depressive symptoms or suicidality. In a study comparing healthy individuals with a family history of MDD and those without, healthy individuals with a family history of MDD showed higher serum BDNF levels, and elevated BDNF levels were suggested as a risk factor for MDD [72]. In a longitudinal study, patients with MDD had a sharper decrease in BDNF levels over time versus HCs [73]. Similarly, changes in BDNF levels were reportedly related to treatment responses. Lee and colleagues found no significant differences in serum BDNF levels between drug-naïve depressed adolescents and HCs. However, they confirmed that the change in BDNF level at 2 weeks of treatment was related to the SSRI response [74]. An early reduction in BDNF level (baseline to week 2) was evaluated as a predictor of the SSRI response at week 8. Taken together, elevated serum BDNF levels may be a finding observed in early-stage adolescent depressive disorder. Further research is required to determine the exact mechanism of action of BDNF.

## 3. Neurohormonal System and Pediatric Depressive Disorder

Many hormonal systems are known to be associated with pediatric depressive disorders, including HPA, hypothalamus-pituitary-gonadal (HPG), hypothalamus-pituitary-somatotropic, and hypothalamic-pituitary-thyroid (HPT) axes. The relationship between these neurohormonal axes and depressive disorders has been established through studies in adults. However, few evidence-based studies in pediatric populations have been performed.

The HPA system is the most studied neurohormonal system in depression. Corticotropin-releasing hormone (CRH), the main activator of the HPA axis, is released in the hypothalamus. CRH reaches the anterior pituitary and stimulates the release of adrenocorticotropic hormone (ACTH), which stimulates cortisol secretion in the adrenal cortex. The HPA axis is known to regulate stress. Depression has been linked to altered stress responses. When we consider the link between youthful depression and stress exposure, efforts to identify related biomarkers involve the HPA axis [75]. 

HPA axis dysregulation has been reported by several studies. Lopez-Duran et al. conducted a meta-analysis of children and adolescents with depression. They analyzed 17 published studies of the HPA axis response to the dexamethasone suppression test (DST) in depressed youth (n = 926), 17 studies on basal HPA axis functioning (n = 1332), four studies that compared cortisol levels after CRH infusion, and three studies that evaluated HPA axis reactivity to psychological stressors in children or adolescents with depression. The HPA axis system tends to be dysregulated in depressed youth, as evidenced by atypical responses to the DST and higher baseline cortisol values. Depressed youths have a normative response to CRH infusion but an overactive response to psychological stressors compared to non-depressed peers [76]. The increased activity of the HPA axis is thought to be related, at least in part, to reduced feedback inhibition by endogenous glucocorticoids—cortisol in humans. Elevated cortisol levels in patients with depression are considered a compensatory mechanism in response to decreased glucocorticoid receptor function and expression in the brain. The disruption of glucocorticoid rhythms may also be related to depression [77].

The relationship between early childhood traumatic exposure and the later development of depression in relation to the HPA axis has been reported [78]. Childhood trauma is related to the sensitization of the neuroendocrine stress response, glucocorticoid resistance—reduced feedback inhibition by endogenous glucocorticoids, increased CRH activity, HPA axis hyperreactivity, and autonomic nervous system hyperreactivity. HPA-axis hyperactivity was also observed in unaffected individuals at familial risk for depression, and it predicted the onset of depression. This suggests that it may be a genetic vulnerability marker for depression [79]. Another study targeting adolescents at high risk for developing depression due to parental depression also showed elevated nocturnal urinary-free cortisol excretion at baseline. Elevated baseline nocturnal urinary-free cortisol levels were associated with the development of depression over a 5-year follow-up period. Thus, we can conclude that these variables showing HPA activity might act as vulnerability biomarkers for depression [80]. This HPA hyperactivity is also related with the inflammatory response system (IRS). Activation of IRS in depression leads to HPA hyperactivity. Pro-inflammatory cytokines induce HPA hyperactivity and activate the nuclear factor kappa-light-chain-enhancer of activated B cells (NF-κB) [81]. NF-κB signaling is known to mediate the suppression of hippocampus neurogenesis caused by stress [82].

One meta-analysis of male depressive disorders is related to the HPG axis. The participants’ ages were 18–97 years. Seventeen studies were included in the analysis. Depressed men showed diminished testosterone and marginally elevated estradiol levels (*p* = 0.055) [83]. Although the influence of the HPG axis in children and adolescents has not been studied to date, the relationship between hypogonadism and depression was suggested in certain studies. The physical changes of puberty are accompanied by psychosocial and emotional changes. Disrupted puberty due to hypogonadism can create a psychological burden along with victimization and bullying that are associated with increased depression severity [84]. Symptoms of hypogonadism may also appear similar to those of depression.

Dorn et al. reported that free thyroxine levels (free T4) [85] and triiodothyronine (T3) uptake [86] were lower in depressed adolescents than in controls. This shows the relationship between the HPT axis and depression. A previous study reported that cortisol levels and TSH levels were significantly elevated in patients versus controls (*p* = <0.001, d = 1.35, large effect size, and *p* = <0.001, d = 0.79, moderate effect size, respectively) in adolescent MDD. These results also show that HPT and HPA axis dysfunction are common in adolescents with MDD. However, no relationship between TSH and cortisol levels was found in depressed adolescents with elevated cortisol levels (97.5th percentile). The interactive functions of both axes are loosely related and differ from the study results of adults. The relationship between the HPT and HPA axes in adolescents is influenced by age-related maturation [87].

## 4. GI Biomarkers and Pediatric Depressive Disorder

As depressive disorders have been shown to be associated with inflammation, the GI tract has emerged as a possible source of such inflammatory activity. Intestinal permeability refers to the flow of material through the wall of the GI tract to the rest of the body. This is how nutrients are absorbed and how potentially harmful substances are prevented from entering the body. Increased intestinal permeability is thought to be a factor in various diseases, including schizophrenia and autism [88,89]. The concept of a “leaky gut” is thought to cause chronic inflammation throughout the body. However, solid evidence of this remains relatively scarce. Nevertheless, gut biomarkers are associated with depressive disorders; however, studies on pediatric patients are uncommon.

Ohlsson et al. proposed that zonulin and the intestinal fatty acid binding protein (I-FABP) might be biomarkers for depressive disorders. Their results showed that levels of I-FABP, a marker of enterocyte damage, were elevated in suicidal patients versus non-suicidal depression patients, whereas levels of zonulin, a protein that modulates intestinal wall junctions, were decreased in suicidal patients [90]. These results suggest a possible relationship between intestinal permeability and depressive symptom severity. Zonulin levels are altered in pediatric patients with autism, attention deficit and hyperactivity disorder, or obsessive compulsive disorder; however, no studies have confirmed changes in children and adolescents with depression [91].

A more recent study was conducted on unmedicated adolescents with MDD. Gut permeability was assessed using mannitol and lactulose. An increase in permeability was measured by a higher proportion of lactulose absorption versus mannitol, since lactulose is a larger disaccharide. This lactulose-to-mannitol ratio (LMR) was significantly associated with depression severity, particularly with neurovegetative symptoms [92].

Another important concept is the gut microbiome, the flora that live in the human digestive tract. Changes in the microbiome are related to increased levels of pro-inflammatory cytokines, which in turn are associated with various psychiatric disorders. A recent review article summarized the effects of the gut microbiome on adolescent mental health. Changes in the microbiome impact the HPA axis and can induce depressive symptoms through vagal nerve stimulation or by activating the kynurenine pathway [93]. Such results are backed up by animal studies in which germ-free mice receiving fecal microbiota from depressed patients showed corresponding symptoms, whereas mice transplanted with microbiota from HCs did not [94]. Researchers have focused on the various strains of bacteria and fungi that might cause depressive symptoms, but microbiome compositions are heavily influenced by diet, medication, infection, and other environmental factors, thereby adding many confounding factors. Commonly mentioned strains belong to the phylum of Bacteroidetes and Firmicutes, although such results are not consistent with those of previous studies [95]. Additionally, the microbiome consists of bacteria and fungi, and the fungal composition of depressed patients is also known to be altered. Jiang et al. showed that during a depressive episode, Candida, Chaetomium, Neocosmospora, Occultifur, Neocosmospora, Clonostachys, and Chaetomium were overrepresented in patients with MDD, whereas Scedoporium, Purpureocillium, Penicillium, Clonostachys, and Aureobasidium were underrepresented [96]. The microbiome of depressed children has been studied to a less extent. Michels et al. gathered children and adolescents aged 8 –16 years and measured their stress levels and sequenced their gut microbiomes. The results showed that high stress was associated with lower Firmicutes abundance and higher Bacteroidetes and Euryarchaeota abundance [97]. Such findings suggest that the gut microbial composition might be a biomarker for pediatric depression, but some researchers suggest that the microbiome is not associated with depression or the use of antidepressants at all [98]. Therefore, it is necessary to identify strains of bacteria and fungi that show higher sensitivity and specificity for depression through more controlled larger scale studies.

The oral microbiome is another focus of recent research. Although evidence is scarce compared to its gut counterpart, a few studies examined the association between the oral microbiome and pediatric depression. Wingfield et al. compared the salivary microbiome of adolescents with MDD and HCs and found that a total of 21 taxa of bacteria were more abundant and 19 others were less abundant in MDD patients. The most abundant genera in the patient group were Prevotella nirescens and Neisseria [99]. Another study focused on the diversity of bacterial taxa, but the results showed that the composition, rather than diversity, mattered when associating with depression or anxiety symptoms in young adults. Related taxa of abundance were Spirochaetaceae, Actinomyces, Treponema, Fusobacterium, and Leptotrichia [100].

Research on the microbiome and relevant GI biomarkers is increasing rapidly, focusing on the gut and other parts of the GI tract. Nevertheless, children and adolescents are not a centrally targeted group because of the natural difficulty of the study design. Further in-depth research is crucial for establishing a standard for the use of GI biomarkers in pediatric patients with depression.

## 5. Oxidative Stress Biomarkers and Pediatric Depressive Disorder

Oxidative stress is the damage induced by reactive oxygen species as a result of insufficient detoxification of reactive intermediates by the biological system. Oxidative stress is thought to be associated with various psychiatric disorders, including autism, attention deficit hyperactivity disorder, and depression.

Reliable markers of oxidative stress on DNA and lipids include 8-hydroxy-2′-deoxyguanosine (8-OHdG) and F2-isoprostanes. A meta-analysis by Black et al. reviewed the association between these biomarkers in depressed patients and reported that both were increased [101]. Interestingly, 8-OHdG levels were strongly associated when measured with plasma or serum samples but not urine samples. Another study on inflammatory and oxidative stress markers and depression by Lindqvist et al. showed that F2-isoprostanes and 8-OHdG were both elevated in depressed patients. This study also showed that patients who did not respond to SSRI treatment had higher F2-isoprostanes levels before and after treatment and higher 8-OHdG levels over the course of treatment [102]. Such studies may be small in number, but they provide possible evidence for the use of oxidative stress biomarkers in depressive disorders. 

Studies on oxidative stress in pediatric depression are limited. A pilot study by Horn et al. of 50 adolescents suggested that elevated F2-isoprostanes levels were associated with internalizing symptom score severity and the existence of four or more ACEs, such as abuse or neglect [103]. It should be noted that the clinical diagnosis of psychiatric disorders was not included in this study. Guney et al. matched 40 pediatric and adolescent patients with anxiety disorders with HCs. Oxidative stress was measured with a colorimetric method using a ferrous ioneo-dianisidine complex, resulting in an oxidation reaction with the venous blood sample. The reaction was expressed and measured as the total oxidative status, which was shown to be higher in children with anxiety disorders than in controls [104]. Another interesting review by Tobore suggested that cigarettes and electronic cigarettes can elevate oxidative stress. This, in turn, may be related to various social maladjustments, including aggressive behavior, impaired cognition, and depressed mood [105]. In an animal study, Moradi-Kor et al. worked with adolescent rats and investigated the effects of stress and various measures to nullify such effects. Stress in adolescence is associated with enhanced anxiety levels and depression in adulthood, as well as elevated oxidative stress markers. Environmental enrichment, exercise, and treatment with Spirulina platensis tend to reduce oxidative damage induced by stress [106]. 

Studies on oxidative stress are common. However, none to date were well-designed with an adolescent patient group with clinically diagnosed depression. Further research is necessary to validate the use of oxidative stress markers for various purposes in pediatric patients with depression.

## 6. Summary and Conclusions

There are many biomarkers for pediatric depression, although the amount of research is not comparable to that of adult depression. The directionality is not consistent in most cases and sometimes even contradictory, although some types of involved cytokines have been repeatedly reported. For example, changes in IL-6 serum levels have been reported as one of the most reproducible abnormalities in MDD, but some studies support the involvement of IL-6 in the pathophysiology of pediatric depression, and others do not [107]. In addition, the directions of alterations in the cytokine levels are not consistent according to the studies. When we consider the number of studies performed in this area, these contradictory results cannot simply be said to be insufficient given the amount of research. This might be due to some factors: (1) different study designs, (2) the heterogeneity in the clinical status of the participants, (3) whether peripheral biomarkers may reflect the central, brain-derived pathophysiology of the pediatric depression, and (4) the possibility that these changes are attributable to indirect, non-specific changes rather than direct and specific associations with key changes in the pathophysiology of pediatric depression.

To avoid problems caused by these factors, we can use machine-learning-driven biomarker discovery, and find additional advanced, clinically useful biomarkers. By using these multiple biomarkers collectively, we can extend the use of these biomarker for more information. If we use biomarkers that are known to be related with the pathophysiology of the pediatric depression with more specificity for the machine learning, we might get more predictive values for the assessment of pediatric depression.

Here, we reviewed various biomarkers, including cytokines, BDNF, neurohormones, GI biomarkers, and oxidative stress indices specifically studied for childhood and adolescent depressive disorders (Figure 1). Although biomarker tests are not yet available as definitive diagnostic tools or to predict treatment prognosis, certain biomarkers have shown significant results in previous studies of pediatric depressive disorder. These findings help us to understand the pathophysiology of pediatric depression better. The biomarkers have the potential for use in diagnosis and prognostic assessment, as well as high-risk-group screening. 

## Figures and Tables

**Figure 1 ijms-22-10051-f001:**
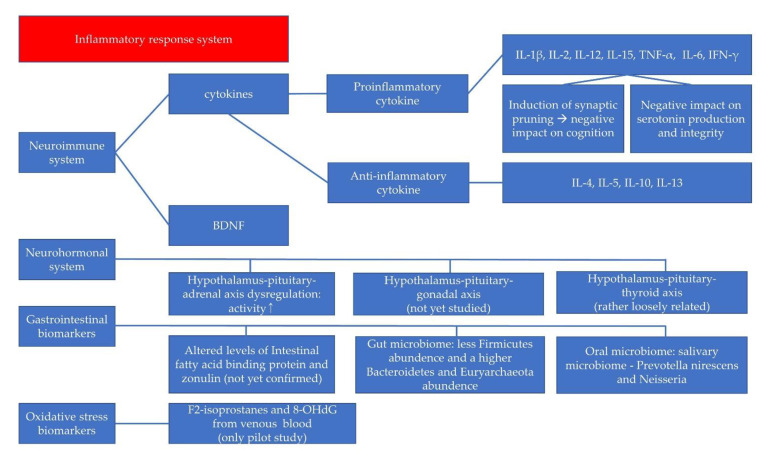
Summary of suggested biomarkers for childhood and adolescent depressive disorders. Pro-inflammatory cytokines induce HPA hyperactivity which is related with the inflammatory response system.

**Table 2 ijms-22-10051-t002:** Previous studies of BDNF in pediatric depressive disorder.

Study	Objective	Design	Findings
Pandey et al, 2010 [21]	To examine the gene expression of BDNF of pediatric depressed Pts (drug naïve or unmedicated for a period of up to 2 weeks)	N: 28, 14 MDD Pts (age 14.9 ± 1.7) and 14 HCs (age 13.0 ± 1.7)	Significantly decreased gene expression of BDNF in the lymphocytes and the protein expression in the platelets of pediatric depressed Pts. compared with HCs
Sasaki et al., 2011 [55]	To investigate whether serum levels of BDNF are altered in pediatric Pts with depression	N: 52, age 8–15 years; 13 male and 17 female MDD Pts; 10 male and 12 female HCs	Significantly lower levels of serum BDNF only in male MDD Pts compared to male HCs, not in female MDD Pts; Significant negative correlation between the serum BDNF levels and the duration of illness in males, but not in females
Pallavi et al., 2013 [56]	(1) To compare serum levels of BDNF in depression patients with healthy controls and (2) to investigate the correlation between clinical severity and serum BDNF levels	N: 148, age 13–18 years; 84 (56 males) MDD Pts, and 64 (39 males) HCs	Adolescents with depression had significantly lower levels of BDNF; BDI-II score showed a statistically significant negative correlation with BDNF in male patients, but not in female patients
Sun et al., 2017 [69]	To investigate (1) the correlation between serum BDNF and depression in children and (2) the change in BDNF after treatment	N: 178, age 7–16 years, 128 (55 males) MDD Pts, and 50 (25 males) HCs	Significantly lower levels of serum BDNF in MDD Pts compared to HCs; MDD Pts with comprehensive nursing showed a significant increase in BDNF expression
Bilgiç et al., 2020 [57]	To identify potential differences in serum BDNF levels in adolescents with MDD compared to HCs	N: 110, age 11–19 years; 70 treatment-free MDD Pts, and 40 HCs	Serum BDNF levels were significantly higher in adolescents with MDD than in HCs; No correlations between the levels of serum BDNF and the severity of depression or suicidality
Lee et al., 2020 [74]	To investigate whether pre-treatment BDNF levels and their early changes predict antidepressant response in MDD Pts	N: 135, age 12–17 years, 83 MDD Pts (46 responders and 37 non-responders), 52 HCs; baseline, 2 weeks, 8 weeks follow-up	No significant findings of serum BDNF between the responders, non-responders, and HCs at baseline; Early decrease in BDNF levels of responders at week 2; Early BDNF decrease predicted later SSRI response at week 8

brain-derived neurotrophic factor (BDNF), self-report Snaith-Hamilton Pleasure Scale (SHAPS), Children’s Depression Rating Scale-Revised (CDRS-R), healthy controls (HCs), patients (Pts), major depressive disorder (MDD), Columbia-Suicide Severity Rating Scale (CSSR-S).

## Data Availability

Not applicable.

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
