# Peer review of "Molecular Biomarkers for Pediatric Depressive Disorders: A Narrative Review"

_ijms, 2021, doi:10.3390/ijms221810051_

Round 1

Reviewer 1 Report

The aim of  of this study was to identify the characteristics of biomarkers in pediatric depressive disorder and  compare the results with biomarkers in adults with depressive disorder. The topic in important. The manuscript is well organized.

Authors discussed four groups of biomarkers in pediatric population:  neuroimmune biomarkers,  neurohormonal biomarkers, gastrointestinal biomarkers and oxidative stress biomarkers and summarized results of previous studies.

Below there are my comments on this manuscript.

In the introduction section authors described problem of depressive disorder, statistic, risks, diagnostic and treatment problems. However, I missed the focus on problem of depressive disorder among children and adolescents and deeper description of challenges and risks. 

In part 2.1. the role of cytokines in neuroinflammation was discussed, studies in pediatric population were presented. My suggestion- to add some thoughts about differences of findings in comparison to adult population.

Authors concluded that biomarkers have to be used for diagnosis and prognostic assessments and for high-risk group screening. Given the abundance of the biomarkers studied, more precise conclusions and recommendations for the pediatric population would still be desirable.

Reviewer 2 Report

The review covers an important topic and there is so much effort put into presenting a considerably informative manuscript. However, it is still in its crude state as it lacks focus.

a) The main aim of the review is not clear. It is still not clear if the reviewed biomarkers are for adults depression or pediatrics?
b) The authors do not explain why they are covering biomarkers for depression in pediatrics. 
c) It  is not clear, what the authors mean by :
"The purpose of this study is to identify the characteristics of biomarkers in pediatric depressive disorders by comparing the results with results from biomarker studies  in adults." 
d) Furthermore, the introduction is about depression in general, but the actual manuscript is about depression in pediatrics. The introduction needs to be re-tuned to cover the topic of the manuscript review.

e) The structure of the review is not consistent. currently the review covers; cytokines,  BDNF,  the relationship between Neurohormonal system and pediatric depressive disorder after a short biological introduction.  Then in the next section, the authors cover the Gastrointestinal (GI) biomarkers and pediatric depressive disorder without any biological introduction. Finally, the authors cover oxidative stress.

f) The section about the role of microbiota is still crude, further details about the bacteria/fungi types and their function and evidence that support their role in depression are still needed. 

g) Overall, the manuscript is vaguely connected.

h) For the review to be informative, more effort will be needed to interpret current knowledge and build of its novel ideas that could lead the field forward. summarizing the existing knowledge might not be sufficient.

 Minor
Perhaps using figures would help the authors decide what are sections they want to expand on.

Round 2

Reviewer 2 Report

I do not have further suggestions.